# Phase Diagram for Ideal Diblock-Copolymer Micelles Compared to Polymerization-Induced Self Assembly

**DOI:** 10.3390/polym12112599

**Published:** 2020-11-05

**Authors:** Alexey A. Gavrilov, Ruslan M. Shupanov, Alexander V. Chertovich

**Affiliations:** 1Physics Department, Lomonosov Moscow State University, 119991 Moscow, Russia; shupanov@polly.phys.msu.ru (R.M.S.); chertov@chph.ras.ru (A.V.C.); 2Semenov Federal Research Center for Chemical Physics, 119991 Moscow, Russia

**Keywords:** diblock copolymers, block-copolymer micelles, polymerization-induced self-assembly, computer simulations, dissipative particle dynamics

## Abstract

In this work we constructed a detailed phase diagram for the solutions of ideal diblock-copolymers and compared such diagram with that obtained during polymerization-induced self-assembly (PISA); a wide range of polymer concentrations as well as chain compositions was studied. As the length of the solvophobic block *n_B_* increases (the length of the solvophilic block *n_A_* was fixed), the transition from spherical micelles to cylinders and further to vesicles (lamellae) occurs. We observed a rather wide transition region between the spherical and cylindrical morphology in which the system contains a mixture of spheres and short cylinders, which appear to be in dynamic equilibrium; the transition between the cylinders and vesicles was found to be rather sharp. Next, upon increasing the polymer concentration in the system, the transition region between the spheres and cylinders shifts towards lower *n_B_*/*n_A_* values; a similar shift but with less magnitude was observed for the transition between the cylinders and vesicles. Such behavior was attributed to the increased number of contacts between the micelles at higher polymer volume concentrations. We also found that the width of the stability region of the cylindrical micelles for small polymer volume concentrations is in good quantitative agreement with the predictions of analytical theory. The obtained phase diagram for PISA was similar to the case of presynthesized diblock copolymer; however, the positions of the transition lines for PISA are slightly shifted towards higher *n_B_*/*n_A_* values in comparison to the presynthesized diblock copolymers, which is more pronounced for the case of the cylinders-to-vesicles transition. We believe that the reason for such behavior is the polydispersity of the core-forming blocks: The presence of the short and long blocks being located at the micelle interface and in its center, respectively, helps to reduce the entropy losses due to the insoluble block stretching, which leads to the increased stability of more curved micelles.

## 1. Introduction

Structured block copolymer systems have attracted great attention due to a large number of application areas, including targeted drug delivery systems, nanoreactors, nanolithography, nanostructured membranes and emulsion stabilizers [1]. However, when using classical diblock copolymers, a complex multi-step procedure (including two-stage synthesis) is required to prepare the system, which also imposes limitations on certain system parameters [1]. In order to overcome these difficulties, the method called polymerization induced self-assembly (PISA) can be used.

The PISA approach is relatively new—the first articles began to appear only recently [2,3,4]. The standard approach in this method is to grow the second block on a pre-synthesized homopolymer precursor; the growing block is solvophobic and tends to precipitate, but the first block, which is solvophilic, stabilizes the micelles. A great advantage of PISA in comparison with the classical method of obtaining micelles from diblock-copolymers prepared in advance is that it is possible to use substantially higher polymer concentrations [1,5]. Two classes of systems can be distinguished: emulsion PISA [2,3,6] and dispersive PISA [7,8,9]. In the former case, the monomer is initially insoluble in the selected solvent, so the polymerization takes place in the droplets enriched with it, whereas in the latter case, the incompatibility between the solvent and the solvophobic monomer is not so large, and only its segments of a certain length are insoluble. In this paper, we are interested in the second case (dispersion PISA), since it allows one to control the polymerization process more precisely. It should be noted that most experimental articles that investigate PISA use the reversible addition–fragmentation chain transfer (RAFT) polymerization [1,4,7,10,11], but there are examples of the use of atom-transfer radical polymerization (ATRP) [12,13,14], nitroxide-mediated polymerization (NMP) [3,6] and even living anionic polymerization [15]. There are already a number of works on applying PISA in various fields [16], including pharmacology [17,18] and even artificial biology [19]. We can mention reviews [5,20] as a fresh and concise viewpoint of the recent advances in PISA. However, despite the high number of works on PISA, not all the properties of this approach are understood, and computer simulations seem to be a perfect tool to perform an in-depth analysis of the mechanisms governing the system behavior.

As for the study of PISA by simulations, there is still no more or less systematic description at the time. In the work [21], different shapes of micelles formed by diblock copolymers obtained by PISA were obtained using the dissipative particle dynamics method. Depending on the length of the macroinitiator, the authors obtained various structures including spherical micelles, cylindrical micelles, vesicles, rings and layers. However, only single concentration was investigated, the lengths of the resulting copolymers were extremely small (the maximum average length was 12), and unrealistically large incompatibility between the solvent and the monomers of growing blocks was used. Later a similar approach was used to model PISA in star [22] and rod-coil [23] block-copolymers. In a recent paper [24] the authors used a realistic experimental system to obtain the simulation parameters, and the structure formation for several chain compositions and reaction speeds was studied. To the best of our knowledge, however, there is no comprehensive study of the phase behavior in such systems, and, what is more important in our opinion, the direct comparison between the micelle formation in solutions of presynthesized diblock copolymers and the micelle formation through PISA within the same model has not been thoroughly conducted.

The most well-known theoretical model describing the morphologies of micelles formed by diblock copolymers is that proposed by Zhulina et al. [25,26,27] in response to the detection of unusual morphologies of the highly asymmetric PS-PAA by Eisenberg [28]. In that model, the self-organization of flexible diblock copolymer chains (both charged and not charged) in the limit of an infinitely dilute solution was considered; the authors explained the sequence of changes in the morphologies (sphere–cylinder–vesicle/lamellae) upon variation of the ratio of the solvophobic/solvophilic parts of the diblock copolymer as well as determined the ratio of the block lengths at which the changes in morphology occur and also compared their results with experiments [26]. See [29] for a recent review of advances and current challenges in the block–copolymer micelles theories. However, the theoretical model developed in [26] does not take into account some of the features typical for PISA. First, this model does not describe the process of self-organization in concentrated solutions, and it is this area that is most interesting not only for experimental research, but also for possible practical applications. Second, the model considers only monodisperse chains, while polydispersity can significantly change the system behavior. Finally, the model works for presynthesized copolymers and does not consider the polymerization process.

In this work, we construct a detailed phase diagram for the solutions of presynthesized diblock-copolymers and compare such diagram with that obtained during PISA. Such comparison allows us to distinguish how the presence of polymerization by itself influences the system properties. A rather general copolymer model was studied, and we deliberately avoided an attempt to study a specific system in order to achieve general understanding of the physical principles of the self-assembly in such systems.

## 2. Method and Model

### 2.1. Method

Simulations were carried out using the well-known dissipative particle dynamics (DPD) method [30,31]. This method is a version of coarse-grained molecular dynamics; the polymer chains are represented by the beads-and-spring model. Beads interacting by a conservative force (repulsion) $F_{ij}^{c}$ a bond stretching force (only for connected beads) $F_{ij}^{b}$, a dissipative force (friction) $F_{ij}^{d}$, and a random force (heat generator) $F_{ij}^{r}$. The total force is given by:(1)$$F_{i}=\underset{i\neq j}{\sum }F_{ij}^{c}+F_{ij}^{b}+F_{ij}^{d}+F_{ij}^{r}$$

The soft-core repulsion between the *i*- and *j*-th beads is equal to:(2)$$F_{ij}^{c}=\{\begin{array}{c} a_{\alpha \beta }\left(1-r_{ij}/R_{c}\right)r_{ij}/r_{ij},\;r_{ij}\leq R_{c}\\ 0,\;r_{ij}>R_{c} \end{array}$$
where ***r_ij_*** is the vector between the *i*-th and *j*-th bead, *a_αβ_* is the repulsion parameter if the particle *i* has the type *α* and the particle *j* has the type *β* and *R_c_* is the cutoff distance. *R_c_* is basically a free parameter depending on the volume of real atoms each bead represents [31]; *R_c_* is usually taken as the length scale, i.e., *R_c_* = 1.

If two beads (*i* and *j*) are connected by a bond, there is also a simple spring force acting on them: (3)$$F_{ij}^{b}=-K\left(r_{ij}-l_{0}\right)\frac{r_{ij}}{r_{ij}}$$
where *K* is the bond stiffness and *l_0_* is the equilibrium bond length. Other constituents of ***F_i_*** are a random force $F_{ij}^{r}$ and a dissipative force $F_{ij}^{d}$, acting as a heat source and medium friction, respectively. Their functional form and parameters (i.e., σ = 3) are taken as in the work of Groot and Warren [31]. Detailed description of the simulation methodology can be found elsewhere [31].

In our calculations we used the following parameters: the DPD number density ρ = 3; the integration timestep Δ*t* = 0.04; the bond length *l* = 0; the bond stiffness *K* = 4.0; and the DPD conservative parameter between alike particles *a_ii_* = 25.0. The incompatibility between species is convenient to characterize using the widely used Flory–Huggins parameter χ; its value was calculated using the common expression $\chi_{\alpha \beta }\approx 0.3∆a_{\alpha \beta }$ obtained in the work [31]. The simulation box size was 80^3^, which contains about 1.5 × 10^6^ DPD beads. For the lowest studied polymer volume fraction of 0.05 we checked the results in a larger box of the size 100^3^ in order to rule out the effect of having too few chains in the system.

### 2.2. Reaction Scheme

In order to simulate the radical polymerization reaction, we used the common Monte-Carlo approach [32,33,34,35,36,37,38]. This method has recently been used to simulate radical copolymerization in bulk [36] and pores [37] as well as emulsion polymerization [38], and the results were found to be in qualitative and even quantitative agreement with the experimental data, which proves that this approach is a good choice for studying PISA. Within this approach, the reaction procedure runs after each *τ_0_* DPD steps; we used *τ_0_ =* 200. The reaction procedure consists of the following stages:A growing chain end *i* is selected at random;The list of all free monomer beads located closer than the reaction radius *R_chem_* from the bead *i* is created. The closest bead *j* is determined, and a bond between the beads *i* and *j* is created with the probability *p*_p_. If the bond is not created, the procedure is repeated with the next closest bead from the list until a bond is created or there are no more unchecked monomers.

These stages are repeated *M* times where *M* is the total number of growing chain ends, so that on average every growing chain is checked once every time we run the reaction procedure. The reaction radius *R_chem_* was chosen to be equal to 1.0, i.e., to the interaction potential cutoff distance *R_c_*.

In this work we did not consider the side reactions such as termination or chain transfer; therefore, the reaction is characterized by single parameter: The propagation probability *p*_p_; its value was chosen to be 0.01, which is small enough to ensure that the growing chains have time to precipitate. For simplicity, the initiation probability was chosen to be equal to *p*_p_.

### 2.3. Model and System

In this work, we compare the behavior of two types of systems: The solutions of presynthesized monodisperse diblock copolymers and PISA. The former system serves as the reference system in some sense since it is close to the theoretical model developed in the work [26]. For both systems, we studied 4 polymer volume fractions (at 100% conversion if PISA is considered): 0.05, 0.1, 0.2 and 0.3; the rest of the system is occupied by solvent. The length of the solvophilic A-block (serving as macroinitiator in PISA) was fixed at *n_A_* = 6 for both presynthesised diblock-copolymers and PISA. The length of the solvophobic B-block varied between *n_B_ =* 12 and 42; for the case of PISA, *n_B_* represents the maximum average B-block length observed at 100% conversion. The solvent was athermal for the solvophilic block (*χ*_AS_ = 0), while the B-monomers as well as B-monomer units had rather high incompatibility with the solvent, *χ*_BS_ = 1.9; for simplicity, the incompatibility between the A and B beads was set to *χ*_AB_ = 1.9 as well. The selected *χ*_BS_-value is not high enough to cause precipitation of the B-monomers at any concentration (i.e., dispersive PISA is studied) since even in the case of 50/50 mixture the critical value of *χ* is equal to 2.0 according to the lattice theory, but B-blocks start to aggregate when they are long enough. In what follows, we use the *n_B_*/*n_A_* parameter to characterize the polymer chains composition.

In order to obtain the equilibrium phase diagram for presynthesized diblock copolymers in an efficient manner, a simple trick was used to prepare the initial state of the systems. All the solvophobic polymer chain ends were initially placed close to the center of the box inside a cubic region of the size ~30^3^ (this size is for φ = 0.1, and it was varied to keep the initial polymer concentration inside more or less constant upon changing φ). This way, the location of the transitions between different types of micelles can be found during reasonably fast runs of ~10^7^ steps since the system basically starts from a precipitated state. After that, we simulated the systems with *n_B_*/*n_A_* values around the found transitions starting from completely homogeneous state (see Figure 1). In this case, due to the rather slow diffusion of even small aggregates as well as the presence of kinetic barriers for aggregates merging, the formation of complex micelles (especially vesicles/lamellae) is very slow and can take up to 2 × 10^8^ steps. However, in all the systems the resulting type of the micelles observed in the system was always the same for both types of the initial system states, indicating that the obtained phase diagram is most likely indeed at equilibrium.

For the case of PISA, a somewhat more complex approach was used. Initially, all the systems had the macroinitiators and monomers homogeneously distributed in the simulation box; when 100% conversion was reached, the simulations were stopped and all the resulting chains were artificially dragged inside a cubic region similarly to the case of presynthesized monodisperse diblock copolymers, and then the simulations from such state were performed. Having determined the positions of the transitions between different types of micelles, we restarted the simulations from the states obtained before the chains were dragged inside a cubic region (i.e., from the states obtained during natural system evolution) only for the *n_B_*/*n_A_* values around the found transitions. As in the case of presynthesized diblock copolymers, we found that after long runs the observed micelles type were the same as after the “forced precipitation” of all chains, indicating that the obtained morphologies are most likely at equilibrium.

## 3. Results and Discussions

### 3.1. Micelle Formation in Solution of Presynthesized Monodisperse Diblock-Copolymers

First of all, we would like to briefly discuss the mechanism of the transformation of the micelle morphology in diblock-copolymer micelles. As has been shown both in experiments and theory [26], a change in the morphology is possible only in the case of crew-cut micelles (i.e., when the thickness of the swollen corona is smaller than the radius of the dense core, which is usually realized for asymmetric copolymer compositions) and is associated with the contribution of the conformational entropy of the solvophobic B-block [26,29]. When the length of the B-clock becomes large enough, the increase of the free energy of the corona upon transition from spheres to cylinders or from cylinders to lamellae (vesicles) becomes smaller than the decrease of the elastic free energy of the core, and the morphology transition occurs. In the opposite case of starlike (or hairy) micelles the conformational entropy of the A-block will always win and force the micelles to remain spherical.

We started our simulations with the case of monodisperse presynthesized diblock-copolymers. We constructed a phase diagram in the (composition–volume fraction) coordinates, and the results are presented in Figure 2; Figure 2 also depicts typical snapshots of the observed morphologies. As expected, the entire diagram is divided into three regions: the region of spherical micelles, the region of cylindrical micelles, and the region of vesicles (lamellae). In general, the location and order of these morphologies is in good agreement with the general concepts and theoretical predictions. Moreover, in the region of low volume fractions, the transition lines become almost vertical, as predicted in theory for the case of dilute solutions.

However, the obtained phase diagram demonstrates two interesting features. The first one is the presence of a transitional coexistence region between the spheres and cylinders. In this region, the system contains a mixture of spheres and short cylinders (“worms”); as the *n_B_*/*n_A_* value increases, the mass fraction of the cylinders as well as their average length gradually grows. Moreover, the micelles mixture seems to be in a dynamic equilibrium as the longer “worms” seem to be constantly breaking into shorter ones (or spheres) and reappearing when shorter ones merge. We assume that the presence of such a region is related to the finite chain length in our simulations, and the width of the transitional region will decrease as it increases. At the same time, we did not observe any intermediate states between the cylinders and vesicles (within our calculation accuracy of *n_B_*/*n_A_* which was equal to 0.166), indicating a much more abrupt transition.

Another feature of the diagram is the behavior at high polymer volume fractions (0.2–0.3): The transition region between the spheres and cylinders shifts towards lower *n_B_*/*n_A_* values. We believe that the explanation for this phenomenon is the increased number of contacts between the micelles; such contacts lead to the compression of the coronal A-blocks which is unfavorable from the point of view of their entropy, and the formation of cylinders in such results in more “free” space in the system and less contacts. The same is observed for the transition between the cylinders and vesicles, but the shift is much less pronounced, which can be attributed to the fact that long cylindrical micelles contact with each other much less than small spherical micelles at the same polymer volume concentration.

### 3.2. Phase Diagram for PISA and Comparison to the Theoretical Predictions

We now study the case of PISA; the obtained phase is shown in Figure 3a. We see that the obtained phase diagram is very similar to that obtained for the case of presynthesized diblock-copolymers (Figure 2); for more detailed analysis we present the comparison of the transition lines in Figure 3b. 

We can see that, despite the same shape of the transition lines, their positions are slightly shifted towards higher *n_B_*/*n_A_* values for PISA, i.e., the transitions occur at larger B-block length. While the observed shift of the transition position between the spheres and cylinders is very small and may be simply attributed to the uncertainties in the definition of its actual location due to the presence of the coexistence region, the transition between cylinders and vesicles/lamellae is rather sharp, and the presence of the shift can be determined with good confidence. To find the reason of such behavior, we investigated the differences between the micelles formation in solutions of presynthesized monodisperse diblock-copolymers and during PISA.

Since, as it was mentioned earlier, we used two different types of chain placement in the simulation box to ensure that we reached the equilibrium morphologies, the only main difference between presynthesized diblock-copolymers and PISA seems to be the presence of polydisperse chains in the latter case. It is worth mentioning that in our model only B-block (core-forming) is polydisperse (since all the solvophilic A-blocks have the same fixed length of 6). Figure 4a presents typical B-block-length distributions observed for spherical and cylindrical micelles as well as vesicles.

We see that the distributions are rather narrow due to our “idealistic” polymerization scheme; the dispersity values of the B-block lie between ~1.03 (for vesicles) and ~1.09 (for spheres). We suppose, however, that even such slight polydispersity is enough to result in the observed changes on the phase diagram. Indeed, the works on the microphase separation in melts of diblock copolymers indicate that the behavior of such systems is greatly influenced by the polydispersity of one or both polydisperse blocks [39,40,41,42,43]: the regions occupied by curved (nonlamellar) phases increase, as does the domain spacing. This can be attributed to the existence of long and short chains [42], which are distributed within the domain in a way that the long chains are situated in the domain center while the short chains occupy the interface [39,40], which reduces the losses in the conformational entropy due to stretching. We believe that the same is true for the micelles: indeed, the polydispersity of the core-forming blocks reduces the entropy losses due to the short and long blocks being located at the micelle interface and in its center, respectively, which leads to increased stability of the more curved micelles. In order to illustrate this, in Figure 4b we present a cross-section of the cylindrical micelle obtained during PISA at *n_B_*/*n_A_* = 5.5 and φ = 0.1 (at these *n_B_*/*n_A_* and φ values vesicle was observed for the presynthesized diblock-copolymers); we clearly see that the center of the micelle is indeed occupied by the long B-blocks, while the short B-blocks are located closer to the interface.

Next, we would like to address the question of the comparison of our results with the results of the analytical theory developed in the work [26]. The presynthesized diblock copolymers at low φ seem to be the closest system to that studied in the theory, which predicted the relative width of the stability region for cylindrical micelles to be equal to $∆n_{B}/n_{A}\approx 0.31n_{B}^{c-\nu }/n_{A}$, where $n_{B}^{c-\nu }/n_{A}$ is the value of $n_{B}/n_{A}$ at the cylinder-to-vesicle transition. The value of $n_{B}^{c-\nu }/n_{A}$ obtained from our simulations for φ=0.05–0.1 was equal to 5.41, therefore, the theory predicts that the transition between the spheres and cylinders should occur at $n_{B}^{s-c}/n_{A}\approx 3.73$. Figure 3b depicts the position of these transitions in comparison with the simulation data; we see a very good agreement in the region of small φ (especially given the presence of rather wide transitional region), while for large φ of 0.2–0.3 the discrepancy becomes rather large due to the transition line not being vertical, which is discussed above.

Finally, we would like to note that, as it was mentioned in Section 2.3, in our simulations of PISA we did not observe any differences in the micelles morphology independent of how we simulate the system time evolution—naturally or with artificial chain “precipitation”—indicating that the obtained morphologies most likely are not kinetically trapped. However, this might not be the case for more complex systems with, for example, crystallizable core-forming blocks, copolymers forming micelles with glassy cores or even very long polymer chains, which could get “frozen” in an intermediate state (i.e., at the $n_{B}/n_{A}$ value corresponding to incomplete conversion) during the polymerization.

## 4. Conclusions

In this work we performed an extensive study of the micelle formation in solutions of presynthesized monodisperse diblock-copolymers as well as during polymerization-induced self-assembly (PISA) by means of DPD simulations. A wide range of polymer concentrations as well as chain compositions was studied in order to construct comprehensive phase diagrams, and special care was taken in order to ensure that the observed morphologies are in equilibrium. To the best of our knowledge, such detailed phase behavior investigation by means of computer simulations has not been reported previously.

First of all, we constructed a phase diagram for monodisperse presynthesized diblock copolymers. As the length of the solvophobic block *n_B_* increases (the length of the solvophilic block *n_A_* was fixed in our simulations), the transition from spherical micelles to cylinders and further to vesicles (lamellae) occurs, which is in good agreement with theoretical predictions and experimental results [1,26,29]. We observed a rather wide transition region between the spherical and cylindrical morphology in which the system contains a mixture of spheres and short cylinders, which appear to be in dynamic equilibrium. We attribute such behavior to the finite chain length in our study. The transition between the cylinders and vesicles, on the other hand, was found to be rather sharp. Next, upon increasing the polymer concentration in the system, the transition region between the spheres and cylinders shifts towards lower *n_B_*/*n_A_* values; a similar shift but with less magnitude was observed for the transition between the cylinders and vesicles. Such behavior can be explained by the increased number of contacts between the micelles at higher polymer volume concentrations; such contacts seem to be unfavorable for the coronal blocks entropy, and the morphology transition helps to reduce the number of contacts. We also found that the width of the stability region of the cylindrical micelles for small polymer volume concentrations is in good quantitative agreement with the predictions of analytical theory [26].

Next, the phase diagram for PISA was constructed. To that end, monodisperse solvophilic macroinitiators were used to grow the insoluble block by means of controlled radical polymerization; we used a model polymerization scheme without side reactions and dormant chains, and the resulting dispersity values Ð of the insoluble B-block were rather small and were typically between ~1.03 (for vesicles) and ~1.09 (for spheres). The obtained phase diagram was similar to the case of presynthesized diblock copolymer; however, the positions of the transition lines for PISA are slightly shifted towards higher *n_B_*/*n_A_* values in comparison to the presynthesized diblock copolymers, which is more pronounced for the case of the cylinders-to-vesicles transition. We believe that the reason for such behavior is the polydispersity of the core-forming blocks: the presence of the short and long blocks being located at the micelle interface and in its center, respectively, helps to reduce the entropy losses due to the insoluble block stretching, which leads to the increased stability of more curved micelles.

Summarizing, we would like to note that the effect of polydispersity is sometimes overlooked, while it can significantly change the system behavior even if the dispersity value is rather low. Therefore, it seems important to thoroughly investigate the influence of the side reactions as well as the presence of dormant chains (i.e., reaction routine closer to the experimental) on the morphology of micelles formed by PISA in order to better understand and predict the experimental data. In addition, the systems with rather high polymer volume concentrations (i.e., concentrated solutions) demonstrate properties distinct from those observed in dilute solutions; considering that the properties of the former systems are studied much less than those of melts and dilute solutions, it would be interesting to investigate the phase behavior of such systems, as this can lead to the appearance of materials with new properties.

## Figures and Tables

**Figure 1 polymers-12-02599-f001:**
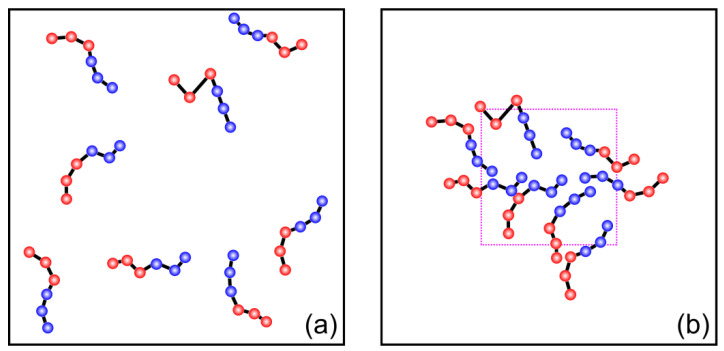
Schematic illustration of two types of initial system states used in this work (**a**) homogeneous initial system state and (**b**) initial system state with the solvophobic chain ends placed inside a cubic region.

**Figure 2 polymers-12-02599-f002:**
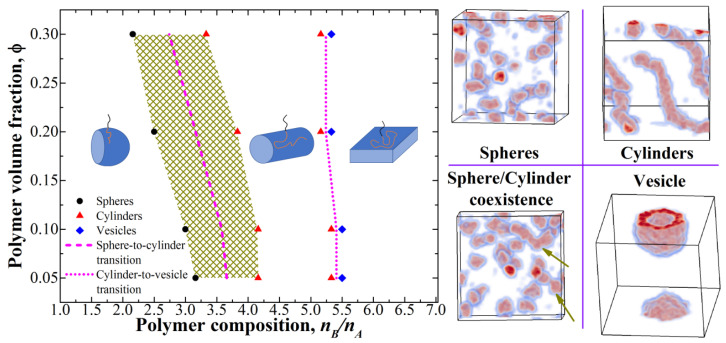
Phase diagram for monodisperse presynthesized diblock-copolymers. Only the points closest to the transitions are shown. The shaded area denotes the transition region between spheres and cylinders; two dark yellow arrows show the “worms” observed in this region in the corresponding snapshot.

**Figure 3 polymers-12-02599-f003:**
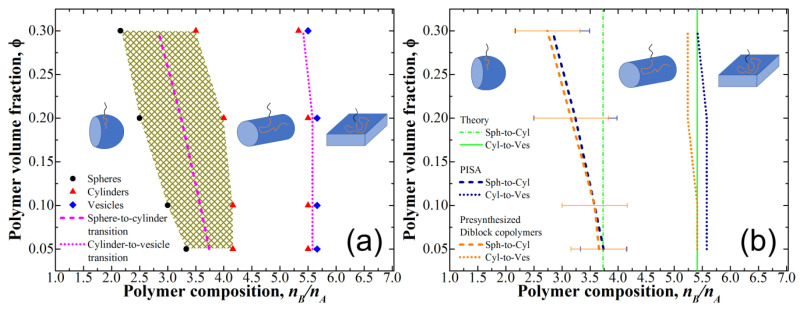
(**a**) Phase diagram for polymerization-induced self-assembly. Only the points closest to the transitions are shown. The shaded area denotes the transition region between the spheres and cylinders; (**b**) Comparison of the transition lines obtained for monodisperse presynthesized diblock-copolymers and polymerization-induced self-assembly (PISA) as well as predictions of the analytical theory [26].

**Figure 4 polymers-12-02599-f004:**
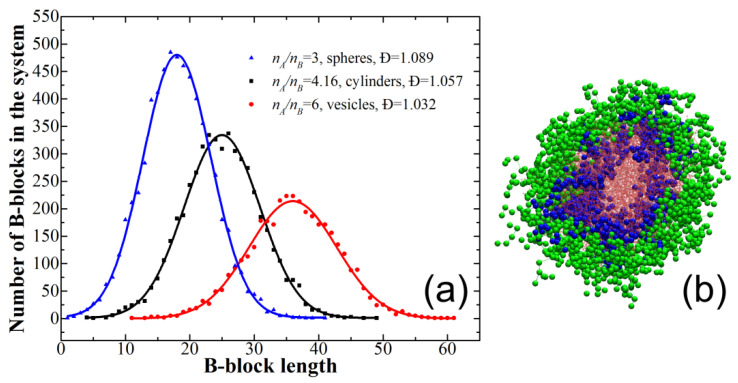
(**a**) Typical solvophobic block-length distributions observed in PISA for φ = 0.1; (**b**) cross-section of the cylindrical micelle obtained during PISA at *n_B_*/*n_A_* = 5.5 (i.e., with average *n_B_* = 33) and φ = 0.1. The solvophilic block is shown in green, B-blocks of length < 27 are shown in blue, and all other B-block (i.e., with a length ≥ 27) are shown in red and are semi-transparent.

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
