# Peer review of "Phase Diagram for Ideal Diblock-Copolymer Micelles Compared to Polymerization-Induced Self Assembly"

_polymers, 2020, doi:10.3390/polym12112599_

Round 1
Reviewer 1 Report
This manuscript describes simulations of phase diagrams of amphiphilic diblock copolymers in selective solvents. The authors compare polymerization-induced self-assembly (PISA) versus phase separation when preformed blocks form the block copolymer. Mainly the transitions from spheres, to cylinders to lamellae have been discussed. Some minor questions remain prior to publication:
- What is the repulsions parameter (delta alpha)? Why is that introduced in addition to the FH chi parameter. Please, explain this to the reader.
- chi(B,S) is rather large. Is that the reason that the micelle core is completely free of the solvent? Please discuss that in the revised manuscript. Why do the authors know that there is no precipitation?
- Can the authors specify the meaning of lamellae/vesicles? In bulk, lamellae occur. But this has no relation to solutions where aggregates occur. Please discuss this or omit the term lamellae.
- The vesicle shown in Figure 1 (right) looks spherical. Do also other types of vesicles appear, e.g. worms?
- I assume that the transition region from spheres to cylinders might show some ellipsoidal structures? Is that the case?
- It is correct that the influence of polydispersity might need separate studies, especially for controlled polymerization techniques.
Author Response
This manuscript describes simulations of phase diagrams of amphiphilic diblock copolymers in selective solvents. The authors compare polymerization-induced self-assembly (PISA) versus phase separation when preformed blocks form the block copolymer. Mainly the transitions from spheres, to cylinders to lamellae have been discussed. Some minor questions remain prior to publication:
We are pleased that the Reviewer positively valued our study.
- What is the repulsions parameter (delta alpha)? Why is that introduced in addition to the FH chi parameter. Please, explain this to the reader.
The parameter Δa is in fact inserted into the DPD model to perform simulations (so it is a technical parameter), while the Flory-Huggins parameter χ is widely used in the polymer community to characterize the incompatibility between species. While in DPD these two parameters are linearly related to each other, for better understanding of the results of our work by the wide audience of the journal we discussed the results in terms of the χ-parameter instead of Δa. We have added a short note to emphasize this to the revised manuscript.
- chi(B,S) is rather large. Is that the reason that the micelle core is completely free of the solvent? Please discuss that in the revised manuscript. Why do the authors know that there is no precipitation?
In our simulations we used χBS=1.9. According to the lattice theory of liquids, the critical value χcrit at which the segregation in a mixture of two liquids occur is equal to χcrit=2.0 (note that this value is for the 50/50 mixture). Therefore, even in the case if only the B-monomer and solvent were present in the system in a 50/50 proportion, we would not observe any segregation between them. In our simulations the fraction of B-monomer in the B-S mixture is much smaller than 50%, so even higher values of χ are necessary to observe segregation. We have added a short note to descibe this to the revised manuscript
- Can the authors specify the meaning of lamellae/vesicles? In bulk, lamellae occur. But this has no relation to solutions where aggregates occur. Please discuss this or omit the term lamellae.
Lamellae is just a planar morphology basically looking like a bilayer. It is worth noting that such morphology is basically equivalent of vesicles and often considered in theory and experiments (see, for example, refs 1 and 26 in the manuscript). The vesicular morphology is simply closed lamellar sheet; this way, there are no edges, but also additional curvature, so usually lamellar sheets close to become vesicles when they are large enough. Therefore, the terms lamellae and vesicle are usually interchangeable in the context of micelles as they feature the same structure of bilayers.
- The vesicle shown in Figure 1 (right) looks spherical. Do also other types of vesicles appear, e.g. worms?
Other shapes of BCP vesicles are usually unfavorable since they would have large solvent-core surface area as well as regions with higher curvature. In our simulations we observed only spherical vesicles.
- I assume that the transition region from spheres to cylinders might show some ellipsoidal structures? Is that the case?
The reviewer is correct that we observed some kind of intermediate structures in the transition region between spheres to cylinders. These structures look like short worms, but they can also be viewed as stretched ellipses.
- It is correct that the influence of polydispersity might need separate studies, especially for controlled polymerization techniques.
We agree with the reviewer that further investigation of the effects of polydispersity is extremely interesting and important task. We plan to address this issue in our future works by studying the influence of the side reactions as well as the presence of dormant chains (i.e. reaction routine closer to the experimental), which we mention in the conclusions.
Reviewer 2 Report
The manuscript Phase Diagram for Ideal Diblock-Copolymer Micelles Compared to Polymerization-Induced Self Assembly, by A.A.Gavrilov, R.M.Shupanov, A.V.Chertovich is well written and describes the thermodynamics and structure of diblock-copolymer micelles, comparing the presynthesized diblock copolymers, the polymerization-induced self assembly (PISA) and the analytical phase diagrams, varying the size of the solvophobic block. There is a quantitative agreement between these phase diagrams, showing regions with spherical, cylindrical and lamellar structures. The methods seem to be adequate and the results are reliable. Although the authors cite properly the literature regarding the methods, a little bit more details in this section (discussion about the dissipative and random forces, a figure with the initial setup, details about the Monte Carlo method) would improve the quality of the manuscript.
Author Response
The manuscript Phase Diagram for Ideal Diblock-Copolymer Micelles Compared to Polymerization-Induced Self Assembly, by A.A.Gavrilov, R.M.Shupanov, A.V.Chertovich is well written and describes the thermodynamics and structure of diblock-copolymer micelles, comparing the presynthesized diblock copolymers, the polymerization-induced self assembly (PISA) and the analytical phase diagrams, varying the size of the solvophobic block. There is a quantitative agreement between these phase diagrams, showing regions with spherical, cylindrical and lamellar structures. The methods seem to be adequate and the results are reliable. Although the authors cite properly the literature regarding the methods, a little bit more details in this section (discussion about the dissipative and random forces, a figure with the initial setup, details about the Monte Carlo method) would improve the quality of the manuscript.
We are pleased that the Reviewer positively valued our study. We have extended the methods section in the revised manuscript.